# The impact of intransitivity on the Elo rating system

**Adam H. Hamilton**\*, **Anna Kalenkova**, **Matthew Roughan**

School of Computer and Mathematical Sciences, The University of Adelaide, Adelaide, South Australia, Australia

\* adam.h.hamilton@adelaide.edu.au

## Abstract

This paper studies the behaviour of the Elo rating system when the underlying modelling assumptions are not met. The Elo rating system is a popular statistical technique used to analyse pairwise comparison data. It is perhaps best known for rating chess players. The crucial assumption behind the Elo rating system is that the probability of which players wins a chess game depends primarily on a real-valued parameter, that quantifies the player's "skill". This implicitly assumes that the binary relation "more likely to win against" is transitive. This paper studies how the Elo rating system behaves when this assumption is relaxed. First, we prove that once the assumption of transitivity is relaxed, the Elo ratings exhibits the undesirable property that estimated ratings are dependent on who plays who. Second, we prove that even when the assumption of transitivity is relaxed, for a given distribution with which players are selected, there is a unique point where the expected change of Elo ratings at that point is zero. This point represents the maximum likelihood estimator of the Elo ratings, given the observed data. Finally, we introduce a statistic that can be used to measure the intransitivity present in a game. We derive this measurement, and demonstrate on simulated data that it satisfies some useful sanity tests.

## 1 Introduction

The Bradley Terry model is a statistical model used to estimate the probability of the outcome of a pairwise comparison. There exist closed form maximum likelihood estimators of the model parameters [1]. In practice, these parameters are often estimated using stochastic gradient descent in a process called the *Elo rating system* [2–5].

The Elo rating system is used to provide numerical ratings for players in many competitive contexts. Elo [6] originally used the system to rate chess players but it has since been applied across a large range of games including many sports and more recently in e-sports, and it has influenced many more recent rating systems, *e.g.,* the Glicko system [7], and the mElo rating system [9].

**Data availability statement:** The data is simulated using Python's PRNG and is available at https://github.com/UofA-AdamHamilton/Elo-Simulations.

**Funding:** Boeing defence Australia top-up scholarship.

**Competing interests:** The authors have declared that no competing interests exist.

Ratings are also of interest to the machine learning (ML) community which is making frequent use of mathematical rating techniques [8,9], for instance in training neural networks.

Whilst very popular, there is still little known about the Elo rating system [17], the importance of theoretical research like this is to place Elo rating system on firm mathematical foundations. This understanding of how Elo ratings behave is what gives us the ability to trust our conclusions drawn from the algorithm.

Most rating systems, including Elo, assume that a player's skill can be encoded by a single number. One implication is that rating systems usually assume skill is transitive. If Alice defeats Bob, and Bob defeats Charlie, then Alice is therefore likely to win against Charlie. But there are many games with some element of non-transitivity, Rock-Paper-Scissors being the canonical example. We focus on the phenomenon of intransitivity because it has largely been neglected in previous research on the Elo rating system. We have not been able to find any mathematically rigorous analysis of how the Elo rating system behaves on data that displays intransitivity and are writing this paper to bridge this existing gap in the literature.

This paper discusses how the Elo rating system behaves in the presence of intransitivity. We show that even when intransitivity is present, Elo exhibits the undesirable property that estimated ratings are dependent on who plays who. More precisely, the long-term outcome of the ratings system doesn't just depend on the skill of the players, it depends on the probabilities with which players are selected to play one another. The parameters to the Elo rating model have often been likened to the Hodge rank of the expected payoff matrix [9]. This dependence on player selection introduces a bias into estimating the expected payoff matrix's Hodge rank.

To clarify, for every set of probabilities $Q$ with which players are selected to play one another, there is a corresponding fixed point in the Elo rating system. The Elo ratings will home in on this point so long as pairs of players are being selected using $Q$. If $Q$ were to change, then so too would the location of this fixed point.

This is surprising. Obviously, the ratings outcomes depend on the particular games played but an implicit belief of those who use Elo and its descendants is that, in the long-term, provided diverse enough games are played, then Elo ratings will converge to meaningful estimates of players' skills. However, if the scores depend on the distribution of games played, then the scores lose meaning. The implication is that ratings in many fields, *e.g.,* sports, e-sports and in training ML algorithms, might not be useful measures. It is hence important to understand why this phenomenon arises so that it can be quantified and perhaps avoided.

Elo ratings can be viewed as a discrete-time Markov chain (DTMC), presuming the outcomes of the games are determined randomly (with probabilities depending on the players' skills). Then the set of Elo ratings for each player forms the Markov chain's state space. With every victory and loss the ratings change depending only on the current ratings and the outcome, with no memory.

Markov Chain Monte Carlo methods such as this abound in statistics [10]. The states visited by the Elo rating system DTMC will tend to be close to the maximum likelihood estimators.

However, this DTMC is hard to analyse because the state space is continuous but not at all smooth. Transitions are binary and result in either an increase or decrease in the players' ratings but the size of these changes is dependent on the current state through a non-linear logistic relationship. As a result, we cannot show that this Markov chain has a stationary distribution, even when the assumptions underlying Elo are true, let alone when they are violated.

However, though we cannot obtain a closed-form expression for the limiting distribution of the DTMC, we can still determine the long-term behaviour of the Elo rating system. In this paper, we do so and use the results to understand how intransitivity affects Elo ratings.

In summary, the contributions (This paper is based on the PhD dissertation [11].) of this work are:

- We characterise the long-term behaviour of the Elo rating system and show that a weaker notion of long-term behaviour (than stationarity) is needed.
- We provide a metric for the degree of non-transitivity in a game and set of players through the *ground-truth advantage matrix A*, which expresses (in ratings-free terms) the advantage/disadvantage of each player over the others.
- We prove in Sect 4 that when *A* is not a *strongly-transitive additive comparison matrix*, then the long-term behaviour of the Elo rating system depends on the probability with which players are selected to play one another.
- We show in Sects 5 and 6, that if the ground-truth advantage matrix and the probability with which players are selected remain constant then there exists a unique *final Elo rating*.

## 2 Background and related work

This paper falls into the now large field of statistical rating and ranking. For introductions see [12] and [6]. There are a large number of approaches to estimating ratings in statistics and data science, but within these, Elo and its variants are perhaps the most commonly used in real ratings.

The *Elo rating system* is a method of estimating the skill of players in a 2-player zero-sum game. It is well known for its use in Chess [13] but has been applied to many other scenarios [14–16]. The Elo rating system works by assuming that every player's skill can be quantified with a scalar number $r$. When two players $i$ and $j$ play one another, the probability that player $i$ defeats player $j$ is given by

$$p_{ij} = \sigma(r_i - r_j),$$

where $\sigma$ is a sigmoid function (typically the logistic function). Then the update rule for ratings given match outcomes implicitly regresses on this relationship. Thus the Elo rating system works both as a model of game outcomes and as a rating system estimation algorithm. There is a large literature concerning the Elo system (for example see [9,13,16,17]) though much of that is not directly relevant here as little of this literature concerns the transitivity assumption implicit in this model.

The Elo rating system itself is a well-studied mathematical object. There are numerous statistical analyses that use the Elo rating system in the field of sports analysis [14,15,18–21], as well as in more novel applications such as assessing fabric quality [22], comparing cyber-security methodologies [16,23], and as a method of determining whether a game is won due to luck or skill [24].

The details of the Elo rating system, and more broadly, the problems of intransitivity in rating systems are also of significant interest to the machine learning community, who are making frequent use of mathematical rating techniques [8,9,34]. In particular, [35] shows that different rating systems can give vastly different results on the same pairwise comparison matrix when intransitivity is present, which is a key motivation for our study.

Some researchers have studied the Elo rating system itself, using techniques from stochastic processes to obtain insights as to how it behaves [4,5,25]. For instance, [3] studies the problem of optimising the parameters used in the Elo rating system, and the bias variance trade-off of the resulting estimators.

Many researchers have noted the limitations of the Elo rating system and have sought to improve it. For instance, [7, 26–29] view the Elo rating system as a hidden Markov model and treat the variances of each player's scores as a variable. Other methods seek to incorporate more information than just wins and losses [2,30–32].

However, underlying all this work, most authors assume that the modelling assumptions (in particular transitivity) of the Elo rating system are satisfied. The papers [9], [33], and [8] are exceptions, and consider the effects of intransitivity on Elo ratings. However, they differ from our research in that they use intransitivity to motivate new rating systems whereas we study the impact of intransitivity on the Elo rating system itself.

Our work focusses on what is called the *final Elo score*. This is the set of numbers to which each player's Elo rating is drawn. Ideally, we would like there to be a single final Elo score for every population of players and we would like this the be the set of skill levels for each player. The Elo rating system was designed as a means to approximate this final Elo score. Our definition of final Elo score matches the one introduced in [9], that claims without proof that the Elo rating system fails when intransitivity is present, the authors do not provide a precise definition of failure. Here we expand on this showing that it succeeds in one sense but fails in another.

More recently [36] noted that when the players are adaptively selected based on their current Elo ratings the ratings may not converge. Our paper explains why this may occur: it is linked to the dependence between the method with which comparisons are selected, and the long-range behaviour of the Elo rating system.

In order to put the behaviour of the Elo rating system on a firm mathematical foundation, we start by studying pairwise comparison matrices [37,38] as well as functions that map these matrices into vectors, *i.e.,* that convert a $m \times m$ comparison of $m$ players into a single rating for each player. To this end, we make use of *combinatorial Hodge theory*, a mathematical description of transitivity and intransitivity in pairwise comparison matrices [35,39–41]. To help explain our methodology, we introduce the following ideas from combinatorial Hodge theory:

The *combinatorial gradient operator* is a function $grad : \mathbb{R}^m \rightarrow \mathbb{R}^{m \times m}$ that maps an $m$-dimensional vector $v$ into a matrix $M$ such that $M_{ij} = v_i - v_j$.

A *strongly transitive additive comparison matrix* (STACM) is a matrix that is an element of the image of *grad*. The set of $m \times m$ STACMs form an ($m$–1)-dimensional subspace of $\mathbb{R}^{m \times m}$.

The *combinatorial divergence function div* : $\mathbb{R}^{m \times m} \rightarrow \mathbb{R}^m$ is a mapping from matrices to vectors given by $div(A) = (A \cdot 1)/m$, *i.e., div(A)* returns a vector whose entries are the arithmetic mean of the corresponding row of $A$.

A *cyclic matrix* is a matrix that belongs to the image of the rotation operator. The rotation operator is a function *rot* : $\mathbb{R}^{m \times m} \rightarrow \mathbb{R}^{m \times m}$, where

$$rot(A)_{ij} = \frac{1}{n} \sum_{k=1}^{m} A_{ij} + A_{jk} + A_{ki}.$$

The set of cyclic matrices form an $(m-2)(m-1)/2$-dimensional vector space over $\mathbb{R}^{m \times m}$.

The set of skew-symmetric matrices $X = -X^T$ form an $m(m-1)/2$-dimensional subspace of $\mathbb{R}^{m \times m}$ and can be decomposed into an orthogonal sum of a STACM and a cyclic matrix [9]. That is, for any skew-symmetric matrix $X$, we can always find a unique STACM $S$ and unique cyclic matrix $A$ such that

$$X = S + A.$$

This is called the *Hodge decomposition* of a skew-symmetric matrix.

## 3 Mathematical formulation

### 3.1 Terminology

Here we state the questions we are trying to answer in a mathematically precise manner. We work with a finite set of $m$ players, where the word "player" is used to denote any entity that can compete (*e.g.,* a human player, team, or an ML solution). The *Elo ratings* are given by a vector, $r \in \mathbb{R}^m$, that assigns a real number to each player. The $i$th entry $r_i$ is called the $i$th player's *Elo rating*.

The *expected payoff matrix* for a game and set of players is a matrix $P \in \mathbb{R}^{m \times m}$, where the entry $P_{ij} \in [0, 1]$, represents the probability that player $i$ will defeat player $j$. In this paper, we do not consider draws, so $P_{ij} + P_{ji} = 1$. In the Elo rating system, we do not directly estimate the expected payoff matrix. Instead, we work with a transformed version called the *advantage matrix*. This matrix is given by $A = \sigma^{-1}(P)$, where $\sigma$ is a sigmoid function – usually the logistic function – applied element-wise to $P$.

The Elo rating system assumes that the probabilities of victory are distributed according to the model above, a variant of the Bradley-Terry model [1]. As players win or lose, their ratings are re-estimated by transferring a portion of their Elo points from the loser to the winner. Victories that are unexpected under the model result in larger transfers of Elo points.

Underlying the model is an advantage matrix, which the Elo rating system assumes to be a STACM, that best explains the observed pairwise comparisons. In the Elo model, the matrix is formed from the combinatorial gradient operator (defined earlier) applied to the rating vector. However, we note that in the case of intransitivity, the idea of estimating the advantage matrix is still valid, though it is no longer a STACM.

We also define the *selection matrix* $Q \in [0, 1]^{m \times m}$ whose entries are the probability that a pair of players will be selected to play one another. It is presumed that such match-ups are randomly chosen according to this matrix, but it is not required that selection be truly random (sequential, weighted round-robin would suffice, for instance). In our work $Q$ is constant, though the results below provide some explanation as to why adaptive choices, *i.e.,* modifying $Q$ in response to outcomes (Adapting match pairings in response to outcomes is not uncommon. One of the main uses for ratings systems is to assign players under the assumption that matches between more evenly rated players will be more interesting for both players and observers.), can result in non-convergent ratings as in [36].

The matrix $Q$ is symmetric since $i$ and $j$ playing one another is the same as $j$ and $i$ playing. We also have that $Q_{ij} \geq 0$, $Q_{ii} = 0$, and $\sum_{i,j=1}^{m} Q_{ij} = 2$, since $Q$ is effectively two identical copies of the same probability mass function. Typically, we would also assume that $Q$ is not decomposable into blocks that don't communicate (more on this assumption will follow).

The core questions of this paper are:

- "Under what circumstances, *i.e.,* for what matrices $P$ and $Q$, does Elo converge to a unique 'final' rating?"
- "Where it exists, how does this final rating depend on the matrices $P$ and $Q$, where naively, one hopes that there is no dependence on $Q$?"

Much of this work is specific to the Elo rating system in particular, but the fundamental idea of understanding the impact of intransitivity on scalar ratings is likely to generalise to Elo's descendants and other (scalar) rating systems that use iterative update rules.

### 3.2 The stability equation

If we assume that $P$ and $Q$ are fixed and that the outcome of every game is statistically independent (conditioned on the players' abilities), then the Elo scores of every player form a discrete-time Markov chain called the *Elo Markov chain*. The Elo ratings of every player at time $t$ are given by the vector $r^{(t)}$. We make the arbitrary choice that every player's rating is initialised at zero, so this DTMC begins at the origin. The long-term behaviour of the Elo rating system can be understood in terms of the limiting distribution of this DTMC.

But this characterisation is not without difficulties. We must take great care when defining the state space of the Elo Markov chain. Because each iteration involves one player giving Elo points to another, the sum of Elo ratings stays constant, the Elo Markov chain is restricted to the subspace $\sum r_i = 0$ which is isomorphic to $\mathbb{R}^{m-1}$. However, this subspace is still a super-set of the Elo Markov chain's state space.

At any given time-step, we select a pair of players using $Q$, and determine the winner using $P$. This means that there are only a finite number of possible values for the vector of Elo ratings at the next time step, $r^{(t+1)}|r^{(t)}$. Since the next vector of Elo ratings can only take one of finitely many values, the state space of the Elo Markov chain is a countable subset of $\mathbb{R}^{m-1}$. Furthermore, it is an open question if any states in this Markov chain are even recurrent. Since we do not know if the Elo Markov chain has any recurrent states, we do not even know if it even has a limiting distribution, let alone have a means to calculate that distribution if it does exist.

**3.2.1 Long-range behaviour.** Since we cannot calculate an explicit limiting distribution of the Elo Markov chain, we focus on a weaker notion of long-term behaviour. Whenever two players Alice and Bob play one another, The player's ratings are updated according to the equation:

$$r_a^{(t+1)} = r_a^{(t)} + \eta\left(S_a - \sigma(r_a^{(t)} - r_b^{(t)})\right),$$

for Alice, and

$$r_b^{(t+1)} = r_b^{(t)} - \eta\left(S_a - \sigma(r_a^{(t)} - r_b^{(t)})\right),$$

for Bob, where $S_a$ is a Bernoulli random variable that equals 1 when Alice wins against Bob, and $\eta > 0$ is an arbitrary gain parameter. The change in Elo ratings is a random variable dependent on the outcome of the game between Alice and Bob. The expected change in Elo ratings is given [17] by

$$r_a^{(t+1)} = r_a^{(t)} + \eta\left(P_{ab} - \sigma(r_a^{(t)} - r_b^{(t)})\right), \tag{1}$$

for Alice, and

$$r_b^{(t+1)} = r_b^{(t)} - \eta\left(P_{ab} - \sigma(r_a^{(t)} - r_b^{(t)})\right), \tag{2}$$

Whenever Alice's rating is too high, then $\eta(P_{ab} - \sigma(r_a^{(t)} - r_b^{(t)}))$ is negative and if Alice's rating is too low, then $\eta(P_{ab} - \sigma(r_a^{(t)} - r_b^{(t)}))$ is positive. This means that, on average, Alice's and Bob's ratings will spend most of their time around values $r_a^{(\cdot)}$ and $r_b^{(\cdot)}$, where $r_a^{(\cdot)} - r_b^{(\cdot)} = \sigma^{-1}(P_{ab})$.

This notion of "spending time close to a value" is what we mean when we talk about the long-term behaviour of the Elo Markov chain. When the quantity $\sigma(r_a^{(t)} - r_b^{(t)})$ is larger than $P_{ab}$, it is expected to decrease. Similarly, when $\sigma(r_a^{(t)} - r_b^{(t)})$ is smaller than $P_{ab}$ it is expected to increase.

There is only one possible value of $r_a^{(t)} - r_b^{(t)}$ for which the Elo Markov chain is neither expected to increase nor decrease. This is where $r_a^{(t)} - r_b^{(t)} = \sigma^{-1}(P_{ab})$. We say that this point is the *final Elo score* for the population $\{a, b\}$. Due to the nature of the Elo Markov chain, we cannot guarantee that this exact set of values is ever achieved or even if such values are possible, but we know that the Elo Markov chain is expected to move towards this value and in the long-term, the Markov chain will be expected to remain close to the final Elo score. This result is enough for practical purposes.

We extend this notion of long-term behaviour to situations with $m$ players in the population. We say that a vector $r$ is a *final Elo score* of the Elo Markov chain, for matrices $P$ and $Q$ iff

$$\mathbb{E}[r^{(t+1)}|r^{(t)}] = r^{(t)}.$$

*i.e.*, the conditionally expected value of the next iteration of the Elo Markov chain at $r$ is unchanged (This property resembles the martingale property, but note that it only holds for one set of values, not the entire space.).

The expected change in the Elo Markov chain is in general given by

$$\mathbb{E}[r^{(t+1)} - r^{(t)}|r^{(t)}] = \sum_j Q_{ij}\eta(P_{ij} - \sigma(r_i^{(t)} - r_j^{(t)})) \tag{3}$$

$$= \eta \, \mathrm{div}\Big( Q \odot (P - \sigma(\mathrm{grad}(r^{(t)}))) \Big), \tag{4}$$

in terms of combinatorial Hodge theory, where $\odot$ is the element-wise (Hadamard) product of two matrices. We solve for the final Elo score of the Markov chain by setting $\mathbb{E}[r^{(t+1)} - r^{(t)}|r^{(t)}]$ to zero and rearranging the expression. This rearranged equation can be written independent of the gain $\eta$:

$$\mathrm{div}(Q \odot P) = \mathrm{div}\big(Q \odot \sigma(\mathrm{grad}(r))\big). \tag{5}$$

We call Eq 5 the *stability equation*.

The stability equation is a necessary condition to determine whether a set of Elo ratings is expected to change, *i.e.*, whether it is a final Elo rating.

## 4 Dependence on the selection matrix

In this section, we will for the moment assume the results from Sect 5, *i.e.*, that for all expected payoff matrices $P$, and all connected selection matrices $Q$, there will always be a unique final Elo score that satisfies the stability equation. We will use this to show how the Elo system can produce different final ratings.

We consider the current result first because it shows why and how the results in Sect 5 are important.

We now present the key theorem of this paper.

**Theorem 4.1.** *If $M = \sigma^{-1}(P) \notin im(grad)$ (i.e., $M$ is not a STACM), then there exist at least two selection matrices $Q_1$ and $Q_2$ such that the final Elo scores associated with $Q_1$ and $Q_2$ are different.*

*Proof*: To prove this, we consider the population of $m$ players as vertices of a graph $G$. The selection matrix $Q$ forms a weighted adjacency matrix which determines how frequently pairs of players play one another. Let us consider the special case where $Q$ is a weighted adjacency matrix of a spanning tree on $m$ vertices. In this case, the stability equation is quite simple to solve. We show an example in Fig 1.

We start by selecting an arbitrary node. In our example in Fig 1, we select the root wlog to be node 1 and assign it a rating of zero, *i.e.*, $r_1 = 0$. For every other node $n$ in the tree, we assign it the rating of

$$r_n = \sum_{(i,j)\in E_{1,n}} \sigma^{-1}(P_{ij}),$$

where $E_{1,n}$ is a path from node 1 to $n$. Since we are dealing with spanning trees, there is a unique path from the root node to any other node in the tree, so each node is assigned a unique rating.

Once we have assigned ratings to each node in the spanning tree, we normalise by subtracting $\sum_i r_i$ from each rating. This results in each node being assigned a rating that satisfies the stability equation such that the sum of ratings is zero.

If $M$ were the combinatorial gradient of a vector, then the sum of edge weights in a path from node $i$ to node $j$ will be the same regardless of the choice of path [40]. This property is due to the fact that $A_{ij} + A_{jk} + A_{ki} = 0$, one of the defining qualities of a STACM [9,40]. If this property does not hold then $M$ is not a STACM.

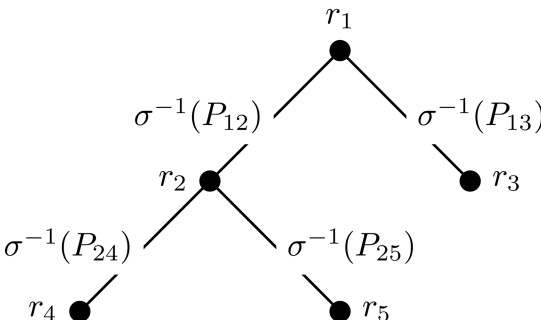

**Fig 1**. **A spanning tree over** $m = 5$ **vertices, where each node represents a player, whose Elo rating is given by** $r_i$. The difference in final Elo score between Player $i$ and $j$ is given by the sum $\sigma^{-1}(P_{i,k_1}) + \sigma^{-1}(P_{k_1,k_2}) + \cdots + \sigma^{-1}(P_{k_n,j})+$, where $k_x$ are the nodes on the unique path from $i$ to $j$. For instance, the path from 4 to 5 goes through node 2, therefore the difference in final Elo scores between Players 4 and 5 is given by $\sigma^{-1}(P_{42})+\sigma^{-1}(P_{25})$.

If there is some non-transitivity present in the game, then the matrix $M$ is not a STACM. Since $M$ is not a STACM then the exist at least two paths between nodes $i$ and $j$ such that the sum of weights along these paths are different. We can select two spanning trees $T_1$ and $T_2$ that contain the first and second paths as subgraphs, respectively.

If we let $Q_1$ and $Q_2$ be weighted adjacency matrices of $T_1$ and $T_2$ then we can solve the stability equations using the aforementioned method. The $r_i - r_j$ will then differ since the path between nodes $i$ and $j$ have different weights, and therefore the final Elo scores will be different. □

This results is a key finding of this paper, namely that the selection matrix can change the final Elo scores. Furthermore, if the matrix $M$ is not a STACM, then the continuity of the stability equation implies that there are uncountably infinitely many values that the final Elo score can take.

Selection matrices $Q$ that correspond to graphs with a single connected component correspond to a unique set of final Elo scores $r$ (we prove uniqueness in Sect 6). This set of potential final Elo scores is called the *Elotope* of a game. The Elotope can be defined as the image of the function that sends a selection matrix $Q$ to its corresponding final Elo score. More details about the geometry of Elotopes can be found in Chapter 3 of [11], where we discuss properties such as openness, boundedness and a polynomial-time algorithm to determine whether a point can be a final Elo score for a given advantage matrix.

The results of this section indicate why the selection matrix has an important effect on the Elo ratings. In the next section, we show that there will be a unique final Elo score when the selection matrix is the adjacency matrix of a more complicated graph (than the trees used here). This shows that even when the ground-truth advantage payoff matrix is not a STACM, the Elo ratings will still find a unique final rating.

## 5 Existence of final Elo ratings

In Sect 4 we assumed that a solution to the stability and conservation equations exists and is unique for a given expected payoff matrix $P$ and selection matrix $Q$. Now we need to show that this is true, which we do in the following theorem. In this section, we prove the first part, *i.e.,* that there exists a solution. Later, in Sect 6, we will examine when that solution is unique.

**Theorem 5.1.** *For a given selection matrix Q, the equation*

$$\text{div}(Q \odot P) = \text{div}(Q \odot \sigma(\text{grad}(r))).$$

*has a unique solution for r if and only if, Q is a weighted adjacency matrix for a graph with one strongly connected component.*

We break the proof into several components, starting with the existence of a solution which is proved using Brouwer's fixed-point theorem.

### 5.1 Conservation and topological constraints

The stability equation, Eq 5, may have infinitely many solutions because the advantage matrix $M$ does not change after adding a constant to all ratings. However, since the sum of Elo ratings stays constant with each iteration, we do not need to worry about this symmetry, and we chose a unique case using the additional constraint

$$\sum_{i=1}^{m} r_i = 0. \tag{6}$$

We call Eq 6 the *conservation equation*.

Another constraint involves the topology of the *interaction network*, an edge-weighted graph whose nodes are players, and whose weighted adjacency matrix is the selection matrix $Q$. The interaction network is required to have exactly one strongly connected component, implying that the matrix cannot be decomposed into blocks. If the interaction network were to have disconnected components, then there would be no information flowing across these components and the players in either component would not be not comparable to each other. Hence, throughout, we assume that the interaction network consists of one strongly connected component.

### 5.2 Existence

In this section, we prove that there exists at least one solution to Eq 5. The key ingredient in our proof is *Brouwer's fixed-point theorem* (FPT), a widely used theorem from topology. There are multiple versions of Brouwer's FPT, of different levels of generality, so we state the particular version that we will use.

**Theorem 5.2** (Brouwer's fixed-point theorem). *Let K be a convex compact subset of a Euclidean space. Every continuous function f : K → K has a* fixed point*, i.e., a point $x^* \in K$ where $f(x^*) = x^*$.*

An interested reader can find a proof of Brouwer's FPT in [42].

We apply Brouwer's FPT to the function $f : \mathbb{R}^{m-1} \to \mathbb{R}^{m-1}$ defining the expected Elo scores at time-step $t + 1$, given Elo scores at time $t$, given by

$$f(r) = \mathbb{E}[r^{(t+1)}|r] = r + \eta \operatorname{div}\left(Q \odot \left(P - \sigma(\operatorname{grad}(r))\right)\right). \tag{7}$$

If $f$ has a fixed point, $f(r^*) = r^*$, then $\eta \operatorname{div}(Q \odot (P_{ij} - \sigma(\operatorname{grad}(r)))) = 0$ has a solution $r^*$, *i.e.,* the stability equation is satisfied and there exists a final score. Hence, our goal is to show that our system satisfies the conditions of the FPT.

Given the conservation equation, we are working in a space isomorphic to $\mathbb{R}^{m-1}$. So, henceforth, we will refer to the subspace $\sum_{i=0}^{m} r_i = 0$ as $\mathbb{R}^{m-1}$.

**Theorem 5.3.** *Given a hypersphere K with sufficiently large radius, R, the conditions of Brouwer's FPT (Theorem 5.2) are satisfied for the function f defined in Eq 7.*

*Proof*: This is a large proof and key parts of the proof are contained in Sects 5.3 and 5.4 as will be indicated.

To start, note that the Elo ratings of every player in the population is a point $r$ in $m$-dimensional space. At each game, we select one of $\binom{m}{2}$ hyperplanes of the form $r_i - r_j = \sigma^{-1}(P_{ij})$. All of these hyperplanes are parallel to the line $(1, \ldots, 1)$ and hence the configuration of hyperplanes has a translation symmetry about this line.

Once we have chosen the hyperplane, the Elo scores of Players $i$ and $j$ will change so that $r$ moves along the null space of $r_i - r_j = \sigma^{-1}(P_{ij})$, and hence moves perpendicularly to the vector $(1, \ldots, 1)$. Therefore at each iteration we will stay in the subspace $\sum_{i=0}^{m} r_i = 0$.

The convex set $K$ that we use in Brouwer's FPT is given by a hyper-sphere of radius $R$ centred at the point $c \in \mathbb{R}^{m-1}$. The purpose of $c$ is to be a point to which we can compare other points on the sphere. The location of the point $c$ can be chosen arbitrarily.

Because $K$ is a hypersphere it is already convex and compact and because $f$ is the composition of continuous functions it is also continuous. The only condition of Brouwer's FPT left to prove is that $f$ maps $K$ into a subset of itself. We do this in two main parts:

1. For sufficiently large $R$, we show the expected change in Elo score maps points from the boundary of $K$ towards its interior.
2. For sufficiently large $R$, we show the expected value of the Elo score cannot leave $K$.

We consider these cases in detail in the two sections below.

### 5.3 Part1

**5.3.1 Step 1.1. Decomposition of $f$.** We start by proving that for sufficiently large $R$, the expected change in Elo score at the boundary of $K$, is directed inwards. For the purposes of this proof, it is useful to think of the Elo rating $r$ as being a point in $\mathbb{R}^{m-1}$ that is moving in discrete time.

Recall that when Player $i$ and $j$ play one another, it is only Player $i$ and $j$'s Elo scores that change and that the expected change in Elo scores for Player $i$ is given by

$$\mathbb{E}[r_i^{(t+1)}|r_i^{(t)}] = r_i^{(t)} + \eta(P_{ij} - \sigma(r_i^{(t)} - r_j^{(t)})).$$

Whenever $i$ and $j$ play one another, the Elo rating $r$, viewed as a point in $\mathbb{R}^{m-1}$, moves in the direction perpendicular to the hyperplane $r_i - r_j = \sigma^{-1}(P_{ij})$. Recall that we update the expected Elo score according to Eq 2. If $P_{ij} - \sigma(r_i^{(t)} - r_j^{(t)})$ is negative, then $r_i^{(t)} - r_j^{(t)}$ will decrease bringing $\sigma(r_i^{(t)} - r_j^{(t)})$ closer to $P_{ij}$. Likewise, if $P_{ij} - \sigma(r_i^{(t)} - r_j^{(t)})$ is positive, then $\sigma(r_i^{(t)} - r_j^{(t)})$ is expected to increase. Either way, the vector of Elo ratings $r$ is expected to move towards the hyperplane $r_i - r_j = \sigma^{-1}(P_{ij})$ rather than away from it. The point $r$ moves perpendicularly to the hyperplane because only $r_i$ and $r_j$ change, and the amount that $r_i$ gains is the amount that $r_j$ loses.

This allows us to express $f(r)$ as a sum of vectors:

$$f(r) - r = \sum_{i \neq j} Q_{ij} v_{ij},$$

where $v_{ij}$ represents the expected change in the vector of Elo ratings $r$ if Players $i$ and $j$ are selected to play one another. The sum is weighted by $Q_{ij}$, which represents the probability with which pairs of players are selected to play each other. Each vector $v_{ij}$ points towards the plane $r_i - r_j = \sigma^{-1}(P_{ij})$ along its null space and $Q_{ij} \in [0, 1)$.

A diagram of $f$, decomposed into a weighted sum of vectors, is shown in Fig 2. We now ask "When is the change in $f(r) - r = \sum_{i \neq j} Q_{ij} v_{ij}$ directed towards the inside of $K$ from its boundary?"

**5.3.2 Step 1.2. When does $v_{ij}$ point inwards?** A simpler question is "when is $v_{ij}$ directed towards the inside of $K$?" A sufficient but non-necessary condition for $f$ to be directed towards $K$'s interior is that all of the weighted vectors $v_{ij}$ are directed towards $K$'s interior. The following lemma specifies when the vector $v_{ij}$ is directed towards $K$'s interior.

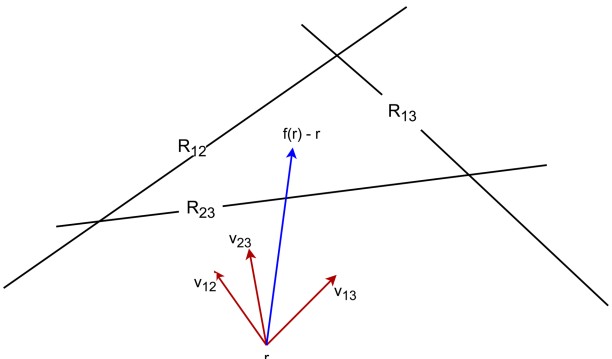

**Fig 2**. An illustration of the average change in Elo score expressed as a weighted sum of vectors $v_{ij}$ along the null spaces of the hyperplanes $R_{ij}$.

**Lemma 5.4.** *Recall that c is the centre of K. Let $R_{ij}$ denote the hyperplane $r_i - r_j = \sigma^{-1}(P_{ij})$, and let $S_{ij}$ denote the hyperplane parallel to $R_{ij}$ that intersects c. Then the vector $v_{ij}$ from any point on the boundary $\delta K$ of K that is in between the hyperplanes $R_{ij}$ and $S_{ij}$ will not be directed towards K's interior, and all other points on $\delta K$ will be directed towards the interior.*

*Proof*: As games involve two players at a time only two Elo ratings will change in any given move, so the change in $r$ is confined to an affine plane containing $r$ and $c$. Moreover $K$ is rotationally symmetric, so it is sufficient to demonstrate the lemma when $m = 3$, *i.e.*, $K$ is a 2-dimensional circle and $R_{ij}$ and $S_{ij}$ are lines.

In this case, Theorem 5.4 can be verified visually – see Fig 3. If $r^{(t)}$ is on the boundary of $K$ in between $S_{ij}$ and $R_{ij}$, then $r^{(t)}$ will be mapped towards $R_{ij}$ pulling it away from $K$'s boundary. For all other points on the boundary of $K$, $r^{(t)}$ will still be pulled towards $R_{ij}$ but in this case, the quickest way to get to $R_{ij}$ is through $K$. Therefore, in this situation, $r^{(t)}$ is mapped to the interior of $K$ (for sufficiently small step sizes). □

We can see that for large $R$ the majority of points on the boundary of $K$ will have $v_{ij}$ directed inwards, but there can be points such that $v_{ij}$ is directed away from $K$. However, $f(r) - r = \sum_{ij} Q_{ij} v_{ij}$ is a (non-negatively) weighted sum of these, so even if $v_{ij}$ is directed away from $K$ for some $i$ and $j$, this may be the exception and the other components of $f(r)$ may be strong enough to ensure that $f(r)$ is directed towards $K$'s interior. This is exactly what we now show.

**5.3.3 Step 1.3. For large R, f is directed towards K's interior.** First we will calculate exactly how much of $Q_{ij} v_{ij}$ is directed towards $K$'s centre and how much is directed away from it. We will also show that as the radius of $K$ approaches infinity, the components of $Q_{ij} v_{ij}$ directed away from $K$'s centre will approach zero for all $i$ and $j$ and the components of $f(r)$ tangential to $K$ and directed towards $K$'s centre will stay above a lower bound. This will complete part one of our proof that the conditions of Brouwer's fixed-point theorem hold.

Let's consider what happens to the vectors $v_{ij}$ when we keep $c$ fixed but let the radius of $K$ approach infinity. As we increase $K$'s radius the distance from $S_{ij}$ and $R_{ij}$ stay the same for all $i$ and $j$. Equivalently, we can view this from the frame of reference of $K$. In this new frame of reference $K$, the hypersphere stays the same size and $R_{ij}$ moves towards $S_{ij}$.

Every point $r$ on the boundary of $K$ will make a unique, acute angle with the hyperplane $S_{ij}$ at $c$, we call this angle $\theta_{ij}$, shown in Fig 4. As the radius of $K$ approaches infinity, $R_{ij}$ and $S_{ij}$ will approach one another and the component of $Q_{ij} v_{ij}$ directed towards $c$ will approach $Q_{ij} v_{ij} \sin(\theta_{ij})$. All other components of $Q_{ij} v_{ij}$ will be tangential to $K$ for all $i$ and $j$. Since $\theta_{ij}$ is acute, $v_{ij} \sin(\theta_{ij})$ will be non-negative.

It is important to note that $\theta_{ij}$ is the limit as the radius of $K$ approaches infinity. Even though $v_{ij} \sin(\theta_{ij})$ is non-negative in the limit as $K$'s radius approaches infinity $f(r)$ may be directed away from the centre of $K$ for any finite radius of $K$.

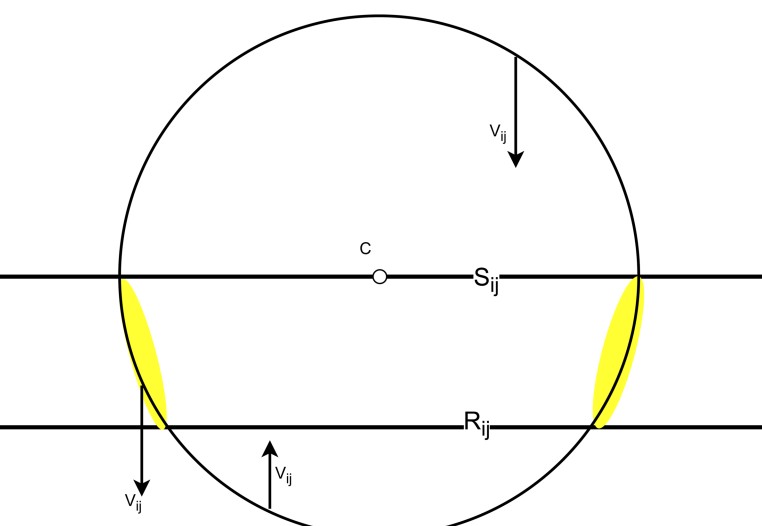

**Fig 3**. **A (2d) hypersphere $K$ centred at $c$.** We draw three arrows from the boundary of the hypersphere to the hyperplane $R_{ij}$, shown below the circle's equator. Two of these arrows are directed to the hypersphere's interior and the third arrow, starting in the yellow region, leaves the hypersphere.

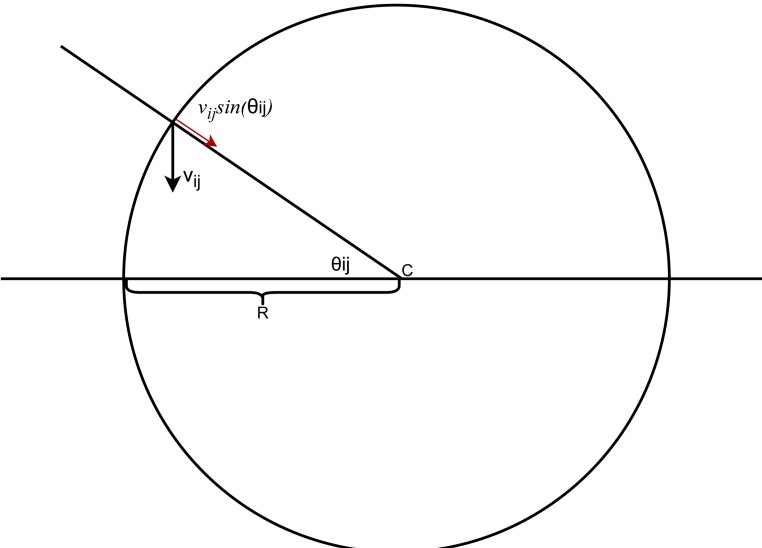

**Fig 4**. **A diagram of $\theta_{ij}$, the unique, acute angle between $r$, $c$, and $S_{ij}$.** The radius of $K$ is denoted by $R$, and the asymptotic component of $v_{ij}$ directed towards the centre of $K$ is given by $Q_{ij}v_{ij}\sin(\theta_{ij})$ as $R$ approaches $\infty$.

Let us consider what happens to $r$, when $r$ is between $S_{ij}$ and $R_{ij}$ for some pair of Players $i$ and $j$. The $Q_{ij}v_{ij}$ component of $r$ will be directed away from $K$ but as $K$'s radius approaches infinity, $r$ will stay between $R_{ij}$ and $S_{ij}$ and $\theta_{ij}$ will approach zero, $v_{ij}\sin(\theta_{ij})$ will approach zero and $Q_{ij}v_{ij}$ will approach a vector tangential to $K$.

When $r$ is not between $R_{ij}$ and $S_{ij}$, $\theta_{ij}$ will not approach zero, hence $v_{ij}\sin(\theta_{ij})$ will be positive and there will be some component of $Q_{ij}v_{ij}$ directed towards $c$, the centre of $K$.

If we have a simply connected interaction network, *i.e.,* a path from every node to every other node, then the $S_{ij}$ hyperplanes will all intersect one another at $c$ and only $c$. This means that there are no points on the boundary of $K$ that belong

to every hyperplane $S_{ij}$ for all $i$ and $j$. Hence, for a sufficiently large $K$, for all points $r$ on $\delta(K)$ there will always be some value of $i$ and $j$ such that $r$ is not between $R_{ij}$ and $S_{ij}$.

If, it were the case that $Q$ did not to form a connected interaction network, then there would exist points on the boundary of $K$ that belong to every hyperplane $S_{ij}$ for all $i$ and $j$, with non-zero $Q_{ij}$. When we have $\binom{m}{2}$ hyperplanes of the form $r_i - r_j = \sigma^{-1}(P_{ij})$ we can find their intersection (if it exists) by solving a system of linear equations. If our interaction network $Q$ is a spanning tree then the corresponding system of linear equations is completely determined and can be solved. If $Q$ is more than a spanning tree, *i.e.,* contains cycles, then we can select all the spanning trees of $Q$ and come up with a finite number of solutions. If there is a finite number of solutions (a bounded set) then we can increase $R$ until $K$ subsumes all of these solutions and there would be no points on the boundary of $K$ that belong to every hyperplane. If $Q$ was not a connected network, then there would be infinitely many solutions and they would form an affine space. Because affine spaces are unbounded, no matter how large we make $K$, it will always intersect every hyperplane. We discuss how the topology of $Q$ relates to systems of linear equations and Elo scores in Sect 4.

The components of all the $Q_{ij}v_{ij}$ terms directed towards $c$ will stay bounded from below since their limit is non-negative, and strictly positive for at least one pair $i$ and $j$. Therefore the sum $\sum_{ij} Q_{ij}v_{ij}$ will have a positive component directed at $c$ and hence $f(r)$ will be directed towards the interior of $K$. The components of $Q_{ij}v_{ij}$ tangential to $K$ will become irrelevant as $K$ approaches infinity.

This completes the first part of our proof that the conditions for Brouwer's FPT hold at the boundary of $K$. Supplementary material that details how large $R$ needs to be for the conditions of Brouwer's fixed-point Theorem to hold is included in section A.

## 5.4 Part 2

Now that we have shown that, for a hypersphere $K$ of sufficiently large radius, then Eq 7 maps elements of $\delta K$ to $K$'s interior, we need to prove that none of the points in the interior of $K$ will be mapped out of $K$.

The distance $f(r)$ can differ from $r$ has an upper bound of $\sqrt{2}\eta$, where $\eta$ is the *learning rate* (or gain) of the Elo updating algorithm, because if Player $i$ and $j$ compete, the expected change to the $i$th coordinate of $r$ is $\eta(P_{ij}-\sigma(r_i-r_j))$, and we subtract the same from the $j$th coordinate of $r$. Since $P_{ij}$ and $\sigma(r_i - r_j)$, are both probabilities, the quantity $\eta(P_{ij} - \sigma(r_i - r_j))$ is between $-\eta$ and $\eta$, therefore for each pair of players we add at most $\eta$ to one coordinate of $r$, and subtract at most $\eta$ from another. Hence $r$ moves at most $\sqrt{2}\eta$ from the original position after a game.

Each pair of players are selected according to $Q$, and since the expected value over a set of vectors of length at most $\sqrt{2}\eta$ must be at most $\sqrt{2}\eta$, $|f(r) - r|$ can be upper bounded by $\sqrt{2}\eta$.

If we take the set $K$ with a sufficiently large radius from Sect 5.3 such that $f(r)$ is directed inwards on $K$'s surface, we can extend the radius of $K$ by an extra $\sqrt{2}\eta$ units. By choosing a radius this long, every point of distance less than $\sqrt{2}\eta$ from $K$'s boundary, will be directed inwards. This follows from part 1. All other vectors $f(r)$ in $K$, will not be long enough to leave $K$.

This completes our proof of Theorem 5.3, Brouwer's fixed point theorem is satisfied and Eq 7 has at least one fixed point. $\qquad\square$

## 6 Uniqueness of the solution

Now that we know that at least one solution exists, we need to show that it is unique. The solutions to the stability equation, Eq 5, represent fixed points of Eq 7. The left hand side of Eq 5 does not depend on the ratings vector, $r$ and hence if we can prove that the right-hand side of the stability equation, $\mathrm{div}(Q \odot \sigma(\mathrm{grad}(r)))$, is an injective function of $r$, then when the stability equation has a solution, it has to be a unique solution.

**Theorem 6.1.** *Given a vector of Elo ratings r for m players, sigmoid function σ, and selection matrix Q, the function*

$$g(r) = div(Q \odot \sigma(grad(r))),$$

*is injective in r if and only if Q is the weighted adjacency matrix of a graph of m vertices with one strongly connected component.*

*Proof*: The first step is to look at the components of *g*. This function consists of the combinatorial gradient operator, which is a linear function from $\mathbb{R}^{m-1}$ to $\mathbb{R}^{\binom{m}{2}}$. After the combinatorial gradient operator, we then apply the non-linear sigmoid function to each element of grad(*r*), multiply the resulting matrix element-wise by *Q*, and finally apply the combinatorial divergence operator.

Geometrically, the function *g* starts with an (*m*–1)-dimensional hyperplane embedded in $\mathbb{R}^m$. The function then applies the combinatorial gradient function resulting in an (*m*–1)-dimensional subspace embedded in $\mathbb{R}^{\binom{m}{2}}$. Then, *g* applies the sigmoid function element-wise to this subspace, mapping it from a flat hyperplane in $\mathbb{R}^{\binom{m}{2}}$ to a curved hypersurface in $(0,1)^{\binom{m}{2}}$.

The second component of *g* consists of projecting the image of $(\sigma(grad(r)))$ back onto the original subspace of $\mathbb{R}^m$. This is done by multiplying every point in the image of $(\sigma(grad(r)))$ element-wise by *Q*, and then applying the combinatorial divergence operator.

For ease of exposition, we will consider the case where *Q* is a selection matrix where every off-diagonal entry is 1 (every diagonal entry is 0 by definition). Once we prove that *g* is injective in this case, we will extend our proof to *Q* where *Q* is the weighted adjacency matrix of a simply-connected interaction network.

The following is a sufficient but not necessary condition for projecting a simply connected, *n*-dimensional, curved hypersurface *X* onto a flat space *S* of dimension $d \geq n$ to be injective: If for all points $x \in X$ such that for all vectors $t \in T_x$, the tangent space of *X* at *x*, *t* does not belong to the null space of *S*, then projecting *X* onto *S* is injective.

We show that there is no point *x* in image of $(\sigma(grad(r)))$, such that $T_x$ contains a vector in the kernel of the combinatorial divergence operator. This will show that when we project the image of $(\sigma(grad(r)))$ onto $\mathbb{R}^m$, no two points will have the same projection.

Because σ is an increasing function, the partial derivative of any entry of grad(*r*) is of the same sign as the corresponding entry in $\sigma(grad(r))$ for any point *r*. Therefore, the vectors in the tangent space of grad will belong to the same orthants as the vectors in the tangent space of σ(grad).

The Hodge decomposition theorem [39] states that the kernel of div is an orthogonal subspace to *im*(grad). Therefore, every vector in the kernel of div belongs to a different orthant to every vector tangent to *im*(grad) and hence every vector tangent to σ(*im*(grad)). Therefore no vectors in *ker*(div) are tangent to *im*(grad) and div(σ(grad(*r*))) is injective when $Q_{ij} = 1$.

Here, we say that two matrices *A* and *B* are orthogonal to one another if $\sum_{i,j} A \odot B = 0$, that is if the sum of the element-wise product of the two matrices is zero. This is equivalent to the typical notion of the dot product in Euclidean space.

Now that we have proved injectivity, for the case where $Q_{i,j} = 1$, we can consider the case involving more general values of $Q_{i,j} \in [0,1]$. In the case where $Q_{ij} \neq 1$, then div(σ(grad(*r*))) is still injective. For any element of *Q*, either $Q_{ij} \in (0,1]$, in which case the tangent vectors of $\sigma(Q \odot grad(r))$ stay in the same orthants, preserving injectivity, or $Q_{ij} = 0$, which projects both σ(*im*(grad)) and *ker*(div) onto a lower dimensional subspace by removing certain coordinates of $\mathbb{R}^{\binom{m}{2}}$. Since this can only move vectors from one orthant to the boundary between two orthants, the tangent vectors of σ(*im*(grad)) will still belong to different orthants than *ker*(div) and injectivity will still be preserved.

We only preserve injectivity when there are enough non-zero entries of *Q*. If enough entries of *Q* are zero then we will be projecting a (*m*–1)-dimensional hypersurface onto a subspace of lower dimension, which cannot be injective.

In this case, "enough" means that $Q$ is the weighted adjacency matrix on $m$ vertices (the number of players with Elo ratings) such that this weighed adjacency matrix has one strongly connected component. In the case where too many entries of $Q$ are zero, then this corresponds to the case where we divide our population of players into multiple groups and prevent any player from players someone from a different group. In this case we would not be able to do any comparisons of players between groups. □

If there was no sigmoid function present in Eq 5, then this would be the same problem that was solved by in [39]. However, the image of ($\sigma(\text{grad}(r))$) has non-zero curvature and the techniques used in [39] do not apply here.

## 7 A measurement of intransitivity

Now that we have concluded the main theoretical results of this paper, we describe a potential application of these results in measuring the degree of intransitivity present in a game.

In Sect 4, we show that the set of final Elo scores for an intransitive game is dependent on the selection matrix $Q$. We call the set of final scores the *Elotope* of a game. Every selection matrix $Q$ corresponds to a point in the Elotope.

In the case where a sub-matrix of the advantage matrix $A$ is a STACM (corresponding to transitive play), then different selection matrices can correspond to the same point in an Elotope. Thus the degree to which the Elotope stretches away from a single point (its size) gives one an intuitive understanding of the level of intransitivity present.

For instance, if the advantage matrix $A$ is a STACM, then the Elotope will be a single point. After one adds an intransitive component to STACM, the Elotope changes from a single point to a connected set that grows as more components of the advantage matrix display intransitivity.

However, if the Elotope is distant from the origin, its size may be less relevant in quantifying intransitivity. When the Elotope is distant there are large Elo ratings present and hence some very low and high probabilities of victory. In this case, the logistic sigmoid function means that the probabilities of victory are quite insensitive to changes in ratings. Hence, we seek a measure of intransitivity that understands when intransitivity is relevant. For instance, a set of three players may display intransitivity, but if the chance of victory of one over the other is near 1 or 0, then the intransitivity will not have an appreciable effect on which players win the majority of games.

Based on these observations, we define a measure of intransitivity $I(P)$, based on the ratio of the Frobenius norm of the transitive component to the Frobenius norm of the cyclic component of the advantage matrix $A$:

$$I(A) = \frac{1 + \|A - \text{grad} \circ \text{div}(A)\|}{1 + \|\text{grad} \circ \text{div}(A)\|}. \tag{8}$$

The norm of the transitive component encodes how far away the Elotope is from the origin. If no intransitivity were present, then the Elotope would be a single point. As this point moves further away from the origin, the difference in final scores between players increases, as does the transitive component of $A$. The cyclic component of $A$ encodes the size of the Elotope. As the size of the cyclic component of $A$ increases, there are gradually more and more possible values that final Elo ratings can take. Because of these facts, the ratio of the cyclic and transitive components of the advantage matrix seems like a logical choice for a measurement of how intransitivity affects the long term behaviour of a game. We add 1 to the denominator to avoid dividing by zero, in the event that there is no transitive component of $A$.

If $I(A)$ is below 1, then this means that the transitive component of $A$ is larger than the cyclic component of $A$ and we say that the advantage matrix is *predominantly transitive*. Similarly, if $I(A)$ is above 1, then we say that our intransitivity measure – Eq 8 – applies the results in our paper, but the method requires a large amount of data for accurate measures. To calculate the cyclic and transitive components of a matrix we need to estimate the advantage matrix $A$, *i.e.,* the probability of victories between every pair of players. One of the main uses of the Elo rating system is to estimate these probabilities with much sparser data because complete datasets are rare. For instance, Elo can estimate the probability of victory between players who have never played. Note though it seems likely that no approach can circumvent the need for

a large dataset because a measure of intransitivity cannot be based on the assumption of transitivity. We cannot use an approach like Elo because this would create a circular chain of reasoning.

This large data cost is one of the main reasons why we have not yet used this measure on any empirical datasets. However, we can show it working on test data, as we do below.

### 7.1 Experimental validation

We demonstrate our measurement on a toy problem. We will consider three players who are playing Rock-Paper-Scissors amongst themselves. We consider two situations. First, the three players can only choose between Rock and Scissors. In this case, we would expect the advantage matrix to be mostly transitive. In the second situation, the three players can choose between Rock, Paper, or Scissors. In this case, we can observe advantage matrices with large cyclic components and thus a larger measurement of intransitivity.

Fig 5 demonstrates our experimental framework visually. We represent the strategy space of Rock-Paper-Scissors as an equilateral triangle. In the first situation, the Rock and Scissors situation, Players 1,2, and 3 have strategies shown on the left edge of the triangle. We look at the measurement of intransitivity as Players 1 and 3 play Rock and Scissors with increasing frequency. In a situation such as this we would expect our measurements to show a predominantly *transitive* set of games.

The second situation involves players starting with Rock, Paper, and Scissors with equal probability and then players 1,2, and 3 start to play Rock, Paper, and Scissors respectively with increasing frequency. In a situation such as this we would expect our measurements to show a predominantly *intransitive* set of games.

For these two situations, we consider different PMFs over the strategy space for each player. From these different strategy PMFs, we can calculate the advantage matrix where the entry $A_{ij}$ is the logit function of the probability that player $i$ beats player $j$. This gives us the ground-truth intransitivity measurement associated with an advantage matrix.

As well as the ground-truth intransitivity measurement, we simulate a sequence of games of Rock-Paper-Scissors between our three players. In our simulations, if there is a draw between two players, then a winner is chosen using a fair coin. This simplifies our analysis, because victory can be treated as a Bernoulli random variable. From these simulated games, we can obtain an empirical advantage matrix, and thus an empirical measurement of intransitivity.

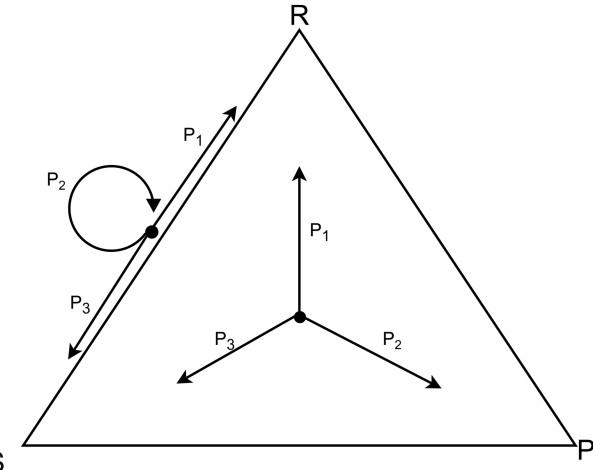

**Fig 5**. **The strategy space of Rock-Paper-Scissors represented as a simplex.** This diagram shows us the two situations considered in our experimental validation. First, we study a predominantly transitive game where the players are restricted to the edge of the simplex between Rock and Scissors. We then study the cyclic game in which Players 1,2, and 3 begin in the centre of the simplex and then move out towards its different vertices.

Specifically, the advantage matrix given player PMFs is given by the equation

$$A = \sigma^{-1}\left(B^T \begin{bmatrix} 0.5 & 0 & 1 \\ 1 & 0.5 & 0 \\ 0 & 1 & 0.5 \end{bmatrix} B\right) \tag{9}$$

where $B$ is the $3 \times m$ matrix whose $i$th column is the probability mass function of Player $i$ playing Rock, Paper, or Scissors respectively.

**7.1.1 Rock and scissors.** We consider three players each of which will play Rock or Scissors according to some probability mass function. We examine the cases where the players' pmfs are given by

$$\begin{bmatrix} 0.5 & 0.5 & 0.5 \\ 0 & 0 & 0 \\ 0.5 & 0.5 & 0.5 \end{bmatrix} + t \times \begin{bmatrix} 1 & 0 & -1 \\ 0 & 0 & 0 \\ -1 & 0 & 1 \end{bmatrix}, \tag{10}$$

where the $i$th column of the matrix is the probability mass function that player $i$ will player Rock Paper, or Scissors respectively, and $t$ is a parameter that ranges from 0 to 0.5. We look at empirically measuring transitivity for three cases with different (probabilistic) strategies (though as only two strategies are used these are all almost transitive):

1. All players play Rock or Scissors with equal probability ($t = 0$).
2. Player 1 plays Rock with probability 0.75, Player 2 players Rock and Scissors with equal probability, and Player 3 plays Rock with probability 0.25 ($t = 0.25$).
3. Player 1 plays Rock with probability 0.9, Player 2 players Rock and Scissors with equal probability, and Player 3 plays Rock with probability 0.1 ($t = 0.4$).

The purpose of these three cases is to look at how the measurement of intransitivity behaves on a game with a predominantly transitive advantage matrix starting from a trivial case, and then look at how this measured intransitivity changes as the difference in players' strategies becomes more extreme. This allows us to examine Eq 8, in a variety of contexts and evaluate against several sanity tests.

In Case 1, all three players play with the exact same mixed strategies. This represents the case where the Player's PMFs have the largest possible entropy. In this case, all players are just as likely to win against one another, therefore the corresponding advantage matrix is the zero matrix, with zero transitive and cyclic components. Therefore the intransitivity measurement will be exactly one. This case is shown in Fig 6 in red, with the ground truth shown by the dashed line, and the empirical estimates from match outcomes shown as the solid line. Note that one is still a comparatively low value for the intransitivity measure.

In Case 2, Players 1 and 3 start to favour Rock and Scissors more of the time. The Shannon entropy of Players 1 and 3 decreases to $H \simeq 0.56$. In this case, the transitive component of the advantage matrix increases more than the cyclic component increases. This case is shown in Fig 6 in blue.

In Case 3, Players 1 and 3 start to favour Rock and Scissors almost all of the time. The Shannon entropy of Players 1 and 3 decreases to $H \simeq 0.056$. This case is shown in Fig 6 in green.

Note that in all of these cases, the measure of intransitivity is small (below one) corresponding to games that are predominantly transitive.

To accompany our empirically measured intransitivity values, we calculate the intransitivity measures for values of $t$ ranging from 0 to 0.5, shown in Fig 7. We can see that, for $t$ close to 0.5, the measurement of intransitivity starts to increase. This is because even though the game of Rock Scissors is very much transitive, it does not completely align

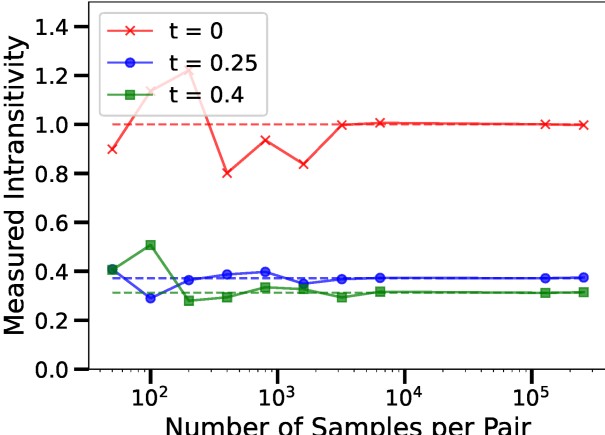

**Fig 6**. **Empirical and ground truth measurements of intransitivity when players can play Rock or Scissors.** We can see that as the size of the datasets increases, the measured intransitivity approaches the ground truth value. We consider three cases where the players choose their strategies with different probability mass functions. We can see that in this setting, for PMFs with both low and high Shannon entropy, the measured intransitivity is relatively low.

with the assumptions made in the Elo model and the advantage matrix is not a STACM. Player 1 will beat Player 2, Player 2 will beat Player 3, and Player 3 will still beat Player 1, but just not with the frequency one would expect by taking the closure of the advantage matrix's spanning tree as we did in Sect 4.

**7.1.2 Rock, Paper, and Scissors.** Now we consider three players each of which will play Rock, Paper or Scissors where the probability mass functions are given by

$$(1-t) \times \begin{bmatrix} 1/3 & 1/3 & 1/3 \\ 1/3 & 1/3 & 1/3 \\ 1/3 & 1/3 & 1/3 \end{bmatrix} + t \times \begin{bmatrix} 1 & 0 & 0 \\ 0 & 1 & 0 \\ 0 & 0 & 1 \end{bmatrix}, \tag{11}$$

where, once again, the $i$th column of the matrix is the probability mass function that player $i$ will player Rock Paper, or Scissors respectively, and $t$ is a parameter that, this time ranges from 0 to 1. Likewise, we look at three cases:

1. All players play Rock, Paper, or Scissors with equal probability of 1/3.
2. Player 1 plays Rock with probability 2/3 and the other two strategies with equal probability. Likewise, Players 2, and 3 choose Paper and Scissors with probability 2/3 respectively and choose from their remaining two strategies with equal probability.
3. Players 1, 2, and 3 choose Rock, Paper, and Scissors respectively with probability 0.9933 and choose equally between the remaining two strategies.

In these three cases, we start where every player is playing at Nash equilibrium, with all strategies being selected with equal probability. We then alter the PMFs of each player so that they favour a single strategy more and more.

In Case 1, shown in Fig 8 in red, all players are equally likely to win. The advantage matrix is the matrix of zeros and the measured intransitivity is 1.

In Case 2, shown in Fig 8 in blue, the players favour particular strategies, and the cyclic component of the advantage matrix increases.

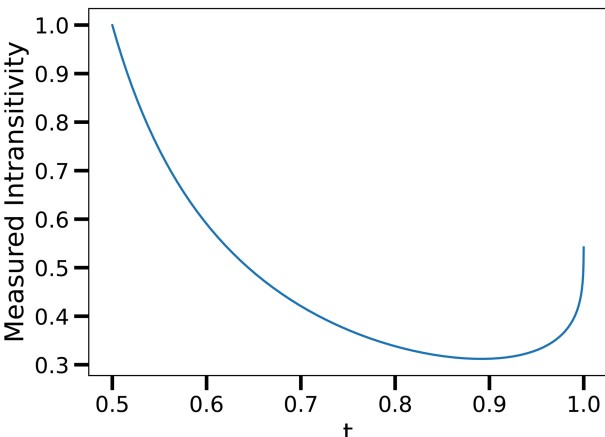

**Fig 7**. **The calculation of measured intransitivity for the Rock Scissors game, where t ranges from 0 to 0.5.** We can see that for all values of *t* considered, the intransitivity is below one. This means that in all cases, the transitive component of the advantage matrix is larger than the cyclic component. We can see that the measured intransitivity starts to increase for larger values of *t*. This is because even though the Rock Scissors game is predominantly transitive, it is not completely transitive in the sense discussed in Sect 4.

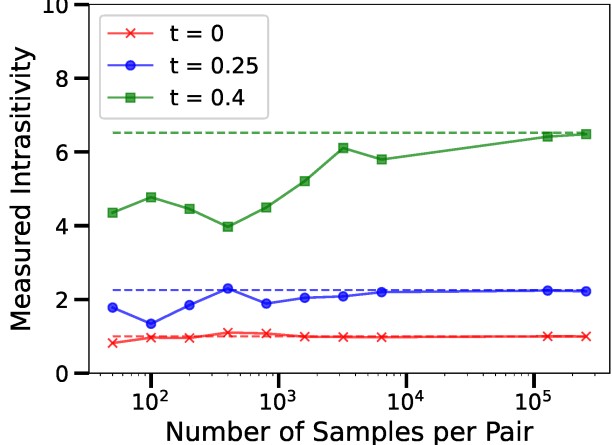

**Fig 8**. **Diagram of our empirical and ground truth measurements of intransitivity, when players can play Rock or Scissors.** We can see that as the size of the datasets increase, the measured intransitivity approaches the ground truth value. We consider three cases where the players play choose their strategies with different probability mass functions. We can see that as the Shannon entropy of each player's PMF decreases, and they start favouring either Rock, Paper, or Scissors more than the others, the measured intransitivity increases.

In Case 3, shown in Fig 8 in green, the players almost exclusively play one strategy. We can see that more samples are required before the empirically measured intransitivity matches the ground truth value. This should not come as a surprise to the reader. Because the players' PMFs are of low Shannon entropy, we do not gain much information with each simulated game.

Likewise, accompanying the empirical measurements is Fig 9, which shows the calculated intransitivity measure as *t* ranges from 0 to 1. One of the main features of this graph is that it is monotonically increasing in *t* and is always showing an intransitivity measurement above one. This is to be expected since, intuitively if each player played Rock, Paper, or Scissors then victory between players would be intransitive. The second noticeable feature of Fig 9 is that it has a vertical

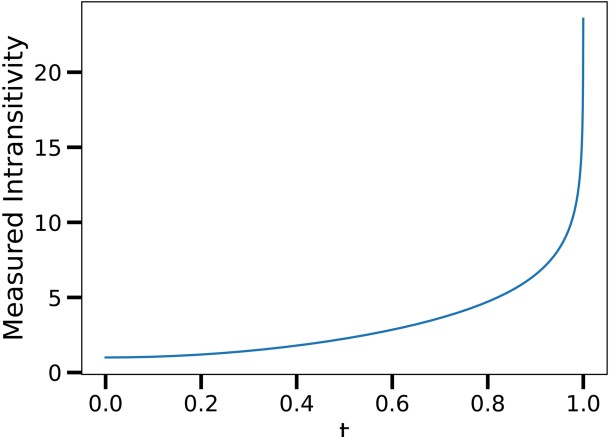

**Fig 9**. **The calculation of measured intransitivity for the Rock-Paper-Scissors game, where _t_ ranges from 0 to 1.** The intransitivity measure is above 1 for all values of _t_ indicating a predominantly intransitive game.

asymptote at $t = 1$. This represents the case where players are playing with pure strategies, therefore the probability one player beats another is either 0 or 1, and the corresponding entry in the advantage matrix is $\infty$ or $-\infty$.

This demonstration shows that our measurement of intransitivity in a game achieves its goal. When the game is mostly transitive, the measured intransitivity is below one and when the advantage matrix has a large cyclic component, then the measured intransitivity will exceed one. In the Rock and Scissors games, the estimated intransitivity measures converged to the ground truth values quicker than their intransitive counterparts. This was because, in the transitive case, the advantage matrix is a lower dimensional object and different data points relating to games between different players all tell a "similar tale". In the intransitive case, the advantage matrix belongs to a higher dimensional space and the entries of the advantage matrix do not relate to one another.

Since we have calculated the intransitivity associated with this version of Rock-Paper-Scissors, it is useful to view this intransitivity using simulations. Here, we take the PMFs of player strategies given in Eq 11, with the parameter $t = 0.85$. We consider two selection matrices

$$Q_0 = \begin{bmatrix} 0 & 0.8 & 0.1 \\ 0.8 & 0 & 0.1 \\ 0.1 & 0.1 & 0 \end{bmatrix},$$

and

$$Q_1 = \begin{bmatrix} 0 & 0.1 & 0.8 \\ 0.1 & 0 & 0.1 \\ 0.8 & 0.1 & 0 \end{bmatrix}.$$

In the case where we sample according to $Q_0$, we have Players 1 and 2 playing one another with high probability. Since Player 1 favours Rock and Player 2 favours Paper, we would expect the Elo ratings to approach values where $r_1 < r_2$. Similarly, since $Q_1$ involves Players 1 and 3 playing one another with increased frequency we would expect final Elo ratings in which $r_3 < r_1$. We simulated the Elo ratings after 1000 games, and plotted the trajectories of these ratings in Fig 10 as well as the Elotope associated with the game. The learning rate of the Elo updates was set to $\eta = 0.02$ to minimise noise and make the different trajectories of the Elo ratings easier to see.

In Fig 10, we can see the trajectories taken by the Elo ratings in the case where players are sampled according to $Q_0$ (shown in blue) and $Q_1$ (shown in orange). We can see that even though the Elo ratings are initialised with the same values, the Elo ratings are drawn to different points of the Elotope. The Elotope of this game is shown in grey in Fig 10.

## 8 Conclusion

This paper considers the long-term behaviour of the Elo rating system in the presence of intransitivity. We demonstrated that notion weaker than the usual Markovian stationarity arguments provides a useful notion of the long-term behaviour of the system.

We were able to show that in the presence of intransitivity Elo can still provide unique solutions. This helps establish that the Elo rating system still "behaves" even when the assumption of a strongly transitive advantage matrix does not completely hold. However, the final Elo score depends on the selection matrix, with different selections leading to different final Elo scores.

Finally, we conclude by discussing a method of quantifying intransitivity in a game. Further questions lie in making this approach more practical and in determining how one might avoid the issues of match dependent final scores.

## A Upper bound on $R$

Here we calculate a rough upper bound to the radius of $R$ such that the conditions for Brouwer's fixed-point theorem hold. We have elected to put this proof in the appendices because it is quite involved and we did not wish to distract from the narrative of the paper. The upper bound calculated in this appendix is only a rough upper bound and when appropriate, we have sacrificed precision for brevity and ease of explanation.

**Theorem A.1.** *There exists a cubic-time algorithm to calculate an upper bound on R such that for all values greater than this upper bound, the conditions for Brouwer's fixed point theorem hold.*

*Proof*: In this proof, we will present the general method of calculating

The trick to calculating this upper bound is to separate the population of $m$ players into groups of three players $\{i, j, k\}$, such that there exist pairs of players $\{i, j\}$ and $\{i, k\}$ such that $q_{i,j}$ and $q_{i,k}$ are both positive.

We calculate a loose upper bound on $R$ in the case that $m = 3$. We then extend this result to larger population sizes by iterating over all possible groups of three players and taking the largest value of these upper bounds. That way, we know

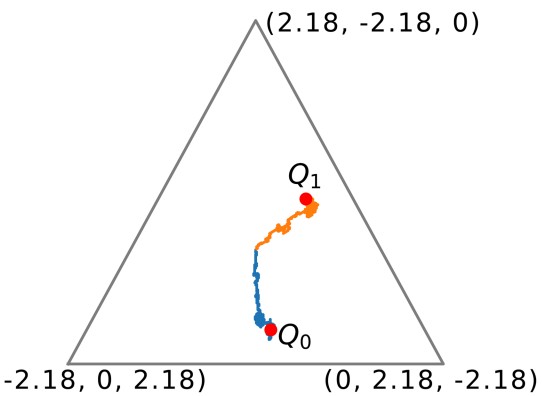

**Fig 10**. **A plot of the trajectories of the Elo ratings of three players playing Rock-Paper-Scissors, with mixed-strategies given by Eq 11, with** $t = 0.85$**, as well as the Elotope associated with the game, shown in gray.** We simulate two trajectories of the Elo rating system, one where players are selected according to the matrix $Q_0$ (shown in blue) and one where players are selected according to the matrix $Q_1$ (shown in orange). We can see that given an intransitive advantage matrix and different selection matrices, the Elo ratings will approach different fixed points.

that $f(r)$–$r$ is directed towards the interior of $K$ for any group of three players in the population. Since this holds for any selected group of three players, it is true for the entire population.

Fig 11 depicts the situation we are dealing with. Here, we have three players $i$, $j$, and $k$. $K$ is a hypersphere, projected onto a circle, with centre $c$. There are three hyperplanes $R_{i,j}$, $R_{i,k}$, and $R_{j,k}$ projected onto lines. These represent the sets points in space where $\sigma(r_i - r_j) = P_{i,j}$. These are shown in Fig 11 in red. It is important to note that these three hyperplanes always intersect each other at an angle of $\pi/3$.

We also have the hyperspace $S_{i,j}$, this is the hyperspace parallel to $R_{i,j}$ but which contains $c$. As was shown in the main body of the paper, the vector $v_{i,j}$ on $\delta K$ is only pointing away from $c$ when it is between $(S_{i,j})$ and $R_{i,j}$. In this proof we will focus on the points $w_1$ and $w_2$. These are the points where $R_{i,j}$ intersects $\delta K$, and thus account for the largest component of $f(r)$–$r$ directed away from $c$. Without loss of generality, we will focus on the point $w_1$. However, we can derive an upper bound to $R$, based on $w_2$ as well and maximise over our two upper bounds for $R$ based on $w_1$ or $w_2$.

Without loss of generality, we will assume that it is $v_{i,j}$ that is directed away from $\delta K$. We will now show that for a sufficiently large radius $R$, even if $v_{i,j}$ is directed away from $K$, the effect from the vector $v_{i,k}$ is enough to bring $f(r)$–$r$ back to pointing to $K$'s interior.

We will begin by upper bounding the component of $v_{i,j}$ that is directed away from $K'$ interior, and we will also derive a lower bound for the components of $v_{i,k}$ that are directed towards the centre of $K$. These pieces of information, will give us an upper bound on $R$ such that $f(r)$–$r$ is directed towards the centre of $K$ when we restrict ourselves to players $i,j,k$.

We start by deriving a lower bound $\lambda(i,j,k,c)$, this is a lower bound to the length of $v_{i,k}$, given $i,j,k$, and $c$, the location of $K$'s centre. Let us consider the hyperplane $R_{i,k}$. This consists of all points $r = (r_1, r_2, \ldots, r_m)$, that satisfy $\sum r_n = 0$, and $r_i - r_k = A_{i,k}$. If players $i$ and $k$ are selected to play one another, then the expected distance travelled by $r$ towards $R_{i,k}$ (and hence the length of $v_{i,k}$ is equal to

$$\eta|\sigma(r_i - r_k) - P_{i,k}|.$$

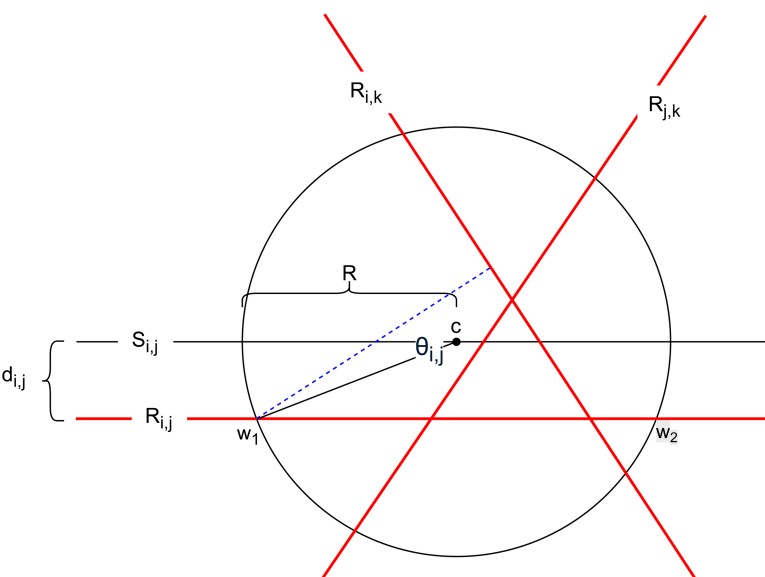

**Fig 11**. **A projection of $K$, with radius $R$ in which we are only considering three players $i$, $j$, and $k$.** The red lines denoted with capital $R$ $R$ represent the hyperplanes that give the true probabilities of victory for a given pair of players.

So the length of $v_{i,k}$ is dependent of $r$'s distance from $R_{i,k}$. As $r$ gets further away from $R_{i,k}$, the length of $v_{i,k}$ approaches either $P_{i,k}$ or $1 - P_{i,k}$. If we look at Fig 11, we can see that $R_{i,k}$ intersects $\delta$ at two points, and as $R$ increases, the distance between these points and $w_1$ and $w_2$ increases as well. Let us define $R_1(\varepsilon)$ as the value, such that for all $R > R_1(\varepsilon)$, $|v_{i,k}| > (1 - \varepsilon)\min(P_{i,k}, 1 - P_{i,k})$.

We fix some value of $\varepsilon > 0$, and let $\lambda(i,j,k,c) = |v_{i,k}| > (1 - \varepsilon)\min(P_{i,k}, 1 - P_{i,k})$. Now that we have our value of $\lambda(i,j,k,c)$, we can find find a lower bound to the component of $f(r)$–$r$ that is directed towards $c$. this is given by

$$\lambda(i,j,k,c)\sin(\pi/3 + \theta_{i,j})\min(Q^+),$$

where $Q^+$ denotes the set of positive valued entries of $Q$.

We need to find an upper bound to the component of $f(r)$–$r$ directed away from $c$. Since the Elo ratings cannot change by more than $\eta$, since the component of $f(r)$–$r$ projected onto the line between $r_1$ and $c$ is given by $v_{i,j}\sin(\theta_{i,j})$ as shown in Fig 11, and since $r$ can belong to at most, $m$–1 regions between some $R_{i,j}$ and $S_{i,j}$, then the component of $f(r)$–$r$ can be upper bounded by the quantity

$$\max(Q^+)(m-1)\eta\sin(\theta_{i,j}).$$

Since we have an under-estimate for the component of $f(r)$–$r$ directed towards $c$ and an over-estimate for the component of $f(r)$–$r$ directed away from $c$, then when

$$\min(Q^+)\lambda(i,j,k,c)\sin(\pi/3 + \theta_{i,j}) - \max(Q^+)(m-1)\eta\sin(\theta_{i,j}) > 0,$$

we will know for sure that $f(r)$–$r$ is pointed towards the interior of $K$.

Using the identities, $\sin(x + y) = \sin(x)\cos(y) + \sin(y)\cos(x)$, and $\sin^2(x) + \cos^2(x) = 1$, we obtain the result that if the following inequality holds:

$$\sin(\theta) < \sqrt{\frac{B^2}{A^2 + B^2}},$$

where

$$A = \min(Q^+)\lambda(i,j,k,c)/2 - \max(Q^+)(m-1)\eta,$$

and

$$B = \min(Q^+)\sqrt{3}\lambda(i,j,k,c)/2,$$

then

$$\min(Q^+)\lambda(i,j,k,c)\sin(\pi/3 + \theta_{i,j}) - \max(Q^+)(m-1)\eta\sin(\theta_{i,j}) > 0,$$

as required.

Since $\sin(\theta_{i,j}) = d(S_{i,j}, R_{i,j})/R_2(i,j,k)$, the distance from $S_{i,j}$ to $R_{i,j}$ divided by the radius of $K$, we can rearrange to obtain

$$R_2(i,j,k) > d(S_{i,j}, R_{i,j})\sqrt{\frac{A^2 + B^2}{B^2}}.$$

Which gives us our desired upper bound on $R$ for a given selection of $i$, $j$, and $k$.

So to summarise, in order to obtain an upper bound on $R$ such that the conditions for Brouwer's fixed point theorem hold.

$$\max_{i,j,k}\Big(\max\big(R_1(i,j,k), R_2(i,j,k)\big)\Big).$$

□

## Author contributions

**Conceptualization:** Adam H. Hamilton.

**Formal analysis:** Adam H. Hamilton.

**Investigation:** Adam H. Hamilton, Anna Kalenkova.

**Methodology:** Adam H. Hamilton.

**Project administration:** Matthew Roughan.

**Software:** Adam H. Hamilton.

**Supervision:** Anna Kalenkova, Matthew Roughan.

**Writing – original draft:** Adam H. Hamilton.

**Writing – review & editing:** Adam H. Hamilton, Anna Kalenkova, Matthew Roughan.

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
