## [Decision Letter · Decision Letter 0]

25 Jul 2025

PONE-D-25-15647The Impact of Intransitivity on the Elo Rating SystemPLOS ONE

Dear Dr. Hamilton,

Thank you for submitting your manuscript to PLOS ONE. After careful consideration, we feel that it has merit but does not fully meet PLOS ONE’s publication criteria as it currently stands. Therefore, we invite you to submit a revised version of the manuscript that addresses the points raised during the review process.

We look forward to receiving your revised manuscript.

Kind regards,

Maryam Afshari

Academic Editor

PLOS ONE

Journal Requirements:

Boeing defence Australia top up scholarship

5. In the online submission form, you indicated that the data is simulated using Pythons PRNG and is available upon request.

6. Please amend either the title on the online submission form (via Edit Submission) or the title in the manuscript so that they are identical.

7. Please ensure that you refer to Figures 4 and 7 in your text as, if accepted, production will need this reference to link the reader to the figures.

Reviewers' comments:

Reviewer's Responses to Questions

**Comments to the Author**

1. Is the manuscript technically sound, and do the data support the conclusions?

Reviewer #1: Partly

Reviewer #2: Yes

Reviewer #3: Yes

2. Has the statistical analysis been performed appropriately and rigorously? 

Reviewer #1: N/A

Reviewer #2: Yes

Reviewer #3: Yes

3. Have the authors made all data underlying the findings in their manuscript fully available?

Reviewer #1: Yes

Reviewer #2: Yes

Reviewer #3: Yes

4. Is the manuscript presented in an intelligible fashion and written in standard English?

Reviewer #1: Yes

Reviewer #2: Yes

Reviewer #3: Yes

5. Review Comments to the Author

Reviewer #1: Thank you for your submission! I have carefully read your article, which focuses on the performance of the Elo rating system when its underlying modeling assumptions are not met, particularly from the perspective of relaxing the transitivity assumption, demonstrating uniqueness. The Elo rating system is a commonly used statistical method for analyzing paired comparison data. This paper proves that after relaxing the transitivity assumption, the Elo rating system exhibits undesirable characteristics where estimated ratings depend on opponents, a finding that deepens the understanding of its limitations. Current research on the Elo rating system is mostly based on conventional assumptions, whereas this paper breaks new ground by exploring more complex and realistic scenarios, offering practical guidance. However, the following issues require your attention:

1. Although the introduced statistical measures were tested on simulated data, the paper lacks integration with real-world applications, missing case studies from games or competitions to further validate their effectiveness and practicality. More in-depth and actionable recommendations grounded in real-world contexts should be provided.

2. The impact of intransitivity on the Elo rating system lacks coverage and consideration of variable factors. Due to the variability of external conditions and influencing factors, the paper fails to present a comprehensive and authoritative perspective, leading to a partial discussion of the Elo rating system's performance. This may overestimate its actual impact.

We understand that completing an academic paper requires extensive effort and significant intellectual dedication. We hope these feedback comments will be helpful for your future research and writing.

Reviewer #2: Review Comments to the Author

The manuscript presents a timely and important investigation into how the Elo rating system behaves under intransitive conditions, an area of growing relevance across various domains including machine learning and performance evaluation.

The theoretical contributions are sound, well-motivated, and clearly articulated. The use of combinatorial Hodge theory to model and measure intransitivity is both novel and appropriate.

The paper is generally intelligible and written in standard academic English, but there are several grammatical and stylistic issues that should be addressed to improve clarity and polish.

Notable language issues include subject-verb disagreement (e.g., “the Elo rating system exhibit” instead of “exhibits”), awkward phrasing, and occasional redundancy (e.g., “this point, this point represents…” in the abstract).

Certain sections, particularly in the introduction and background, could be more concise. Repetition of ideas sometimes disrupts the flow and can be edited for clarity.

Minor formatting inconsistencies are present, likely due to LaTeX typesetting or PDF conversion. These include irregular footnotes, inconsistent citation styles, and occasional spacing issues.

Some definitions and theoretical terms (such as “Elotope”) would benefit from more intuitive explanations or additional context to help non-specialist readers engage more effectively with the material.

Despite these issues, the manuscript offers a meaningful and rigorous contribution to the study of rating systems. The results are valuable and practically relevant.

I recommend a careful round of language and formatting revisions. Once these are addressed, the manuscript will be well positioned for publication.

Reviewer #3: Review Report for PONE-D-25-15647

"The Impact of Intransitivity on the Elo Rating System"

Overall Assessment

This paper makes valuable theoretical contributions to understanding Elo rating systems under intransitive conditions, particularly by providing new insights into the existence and uniqueness of equilibria, established through the application of Brouwer’s theorem. Additionally, the analysis of the dependence on match-selection matrices marks a significant advance in the field. The introduction of the intransitivity metric, I(A), offers a promising new approach for evaluating ranking systems under conditions where traditional transitivity assumptions no longer hold.

The contributions of this paper are especially important for ensuring fairness in competition, as they provide a more nuanced understanding of how rankings can be influenced by non-transitive relationships between competitors. By addressing the complexities that arise from intransitivity, the proposed framework allows for a more accurate and equitable representation of player abilities, ultimately fostering greater fairness in competitive environments.

Below are suggestions aimed at strengthening clarity, empirical validation, and the broader impact of these findings.

Major Comments

1.Clarification of Seemingly Contradictory Contributions

The paper presents two main results:

(i) Ratings depend on match selection (Q) under intransitivity (Section 4),

(ii) A unique equilibrium exists for any fixed Q (Sections 5–6).

At first glance, these findings may seem contradictory.

Suggestion:To address this, explicitly reconcile the two results in the abstract/introduction (e.g., "Although the final ratings are highly dependent on the match-selection distribution Q, we demonstrate that for any fixed Q, a unique equilibrium exists").

2.Theoretical Proofs: Opportunities for Refinement

Theorem 5.3: The condition requiring the hypersphere radius ( R ) to be "sufficiently large" lacks precise quantitative bounds.

Suggestion: Clarify how ( R ) depends on ( \eta ), ( Q ), or ( P ) (e.g., using Lemma 5.4) to enhance the rigor of the proof.

Theorem 6.1: The proof of injectivity assumes ( Q_{ij} = 1 ) for simplicity, but it overlooks more general cases of ( Q ) (p. 21). A more thorough discussion is needed regarding how the curvature of ( \sigma(\cdot) ) impacts injectivity.

Suggestion: Include a remark on how nonlinearity affects orthant preservation, or reference differential geometry tools (e.g., transversality) for a deeper analysis.

3.Empirical Validation Gap

The experiments primarily validate ( I(A) ) (Section 7), but they fail to demonstrate the fundamental phenomenon: match-dependent ratings under intransitivity (Section 4).

Recommendation: Include a straightforward case study (e.g., 3-player cyclic wins) that illustrates how different ( Q ) (e.g., chain vs. star match patterns) result in varying Elo ratings. This would provide empirical support for Theorem 4.1.

Minor Comments

1.Introduction: This section introduces the Bradley-Terry model and its connection to the Elo rating system, explaining how the Elo system is applied in various competitive contexts. While the background and history of the Elo system are clearly presented, the explanation could place more emphasis on why the study specifically focuses on intransitivity phenomena. It would also be helpful to highlight the practical implications for fields like eSports, sports events, and machine learning. While the theoretical framework is well outlined, providing a more intuitive explanation of intransitivity and briefly mentioning real-world applications (such as in competitive games or sports rankings) would strengthen the argument. Overall, the introduction sets up the research well, but simplifying the language and making clearer connections to practical uses would enhance its persuasiveness.

2.Terminology and Notation

The term "Elotope" (p. 15) is introduced without a formal definition.

Suggestion: Provide a definition early on, for example, “the set of all possible final ratings under varying Q.”

There is inconsistent notation, such as grad(r) vs. ∇r (e.g., Eq 4–5), and σ(⋅) vs. logit(⋅).

Suggestion: Standardize the notation to ∇r and σ(⋅) throughout the paper.

3.Figures and Accessibility

Figures 6–9: Axes are unlabeled (e.g., "Number of Samples" in Fig 6).

Figure 5: Simplex vertices should label strategies (R/P/S).

Suggestion: Label axes clearly in all figures and add strategy labels to Fig 5.

Strengths

Theoretical Rigor: The application of Brouwer’s theorem and Hodge decomposition is elegant.

Novel Metric: I(A) cleverly utilizes cyclic/transitive components.

Impactful Rebuttal: The paper effectively addresses Balduzzi et al. [9] by showing partial Elo robustness (p. 11).

Suggested Revisions Summary

Clarify the relationship between match-dependence and equilibrium uniqueness.

Tighten the hyperradius bounds in Theorem 5.3 and discuss the impact of σ(⋅) in Theorem 6.1.

Provide an empirical demonstration of Q-dependent ratings.

Standardize notation and define "Elotope" upon first use.

Improve figure labels and share the simulation code.

This work significantly contributes to rating system theory, and with minor adjustments, its impact will be even greater. I commend the authors for their rigorous analysis and look forward to their response to these suggestions.

6. PLOS authors have the option to publish the peer review history of their article (what does this mean?). If published, this will include your full peer review and any attached files.

Reviewer #1: No

Reviewer #2: **Yes: **Abraham Loha Anebo(PhD)

Reviewer #3: **Yes: **Luhua Xie

---

## [Author Response · Author response to Decision Letter 1]

7 Sep 2025

All responses to reviewer comments have been included in the file Elo_PLOS_One_response_to_reviewers.pdf

---

## [Decision Letter · Decision Letter 1]

20 Nov 2025

The Impact of Intransitivity on the Elo Rating System

PONE-D-25-15647R1

Dear Dr. Hamilton,

We’re pleased to inform you that your manuscript has been judged scientifically suitable for publication and will be formally accepted for publication once it meets all outstanding technical requirements.

Kind regards,

Maryam Afshari

Academic Editor

PLOS ONE

Additional Editor Comments (optional):

Reviewers' comments:

Reviewer's Responses to Questions

**Comments to the Author**

1. If the authors have adequately addressed your comments raised in a previous round of review and you feel that this manuscript is now acceptable for publication, you may indicate that here to bypass the “Comments to the Author” section, enter your conflict of interest statement in the “Confidential to Editor” section, and submit your "Accept" recommendation.

Reviewer #1: All comments have been addressed

2. Is the manuscript technically sound, and do the data support the conclusions?

Reviewer #1: Partly

3. Has the statistical analysis been performed appropriately and rigorously? 

Reviewer #1: Yes

4. Have the authors made all data underlying the findings in their manuscript fully available?

Reviewer #1: No

5. Is the manuscript presented in an intelligible fashion and written in standard English?

Reviewer #1: Yes

6. Review Comments to the Author

Reviewer #1: Thank you for your submission! I have carefully read your article. This paper points out the neglected 'non-transitivity' issue in the Elo rating system, challenging the traditional assumption of 'transitive win-loss relationships' in the system. It fills the gap in understanding the impact of non-transitivity on the Elo system, providing a new perspective for theoretical research on the Elo rating system and is significant for understanding the limitations of rating systems in non-transitive scenarios. The paper rigorously proves the core theorem using mathematical tools such as combinatorial Hodge theory and Brouwer's fixed-point theorem, enhancing the scientific and credibility of the research. However, you still need to review the following issues:

1. The empirical validation of this paper is based solely on the simplified game scenario of 'rock-paper-scissors,' without involving complex data from real sports events or e-sports, which may limit the applicability of the research conclusions in real scenarios.

2. The paper only points out that non-transitivity leads to the Elo rating's dependence on the choice matrix but does not further study how to optimize the Elo system to reduce the impact of non-transitivity, such as improving the rating update algorithm or adjusting model assumptions, making the research application extension insufficient and lacking a solution path for practical problems.

3. The sample size and parameter considerations are limited. The parameter settings for simulated data are relatively fixed, without fully exploring the stability and change patterns of non-transitivity measurement under different parameters and larger sample sizes, which may affect the robustness of the research conclusions.

I sincerely appreciate the time and effort you have invested in writing the paper and hope that the result of this submission will not discourage you from submitting future manuscripts.

7. PLOS authors have the option to publish the peer review history of their article (what does this mean?). If published, this will include your full peer review and any attached files.

Reviewer #1: No

---

## [Editor Report · Acceptance letter]

PONE-D-25-15647R1

PLOS ONE

Dear Dr. Hamilton,

I'm pleased to inform you that your manuscript has been deemed suitable for publication in PLOS ONE. Congratulations! Your manuscript is now being handed over to our production team.

Kind regards,

on behalf of

Dr. Maryam Afshari

Academic Editor

PLOS ONE